# Exploring the Role of Federated Data Spaces in Implementing Twin Transition within Manufacturing Ecosystems

**DOI:** 10.3390/s23094315

**Published:** 2023-04-27

**Authors:** Marko Jurmu, Ilkka Niskanen, Atte Kinnula, Jukka Kääriäinen, Markus Ylikerälä, Pauli Räsänen, Tuomo Tuikka

**Affiliations:** VTT Technical Research Centre of Finland Ltd., Kaitoväylä 1, FI-90570 Oulu, Finland

**Keywords:** sustainability, manufacturing, ecosystem, data sharing, data sovereignty, federated data space, design science

## Abstract

Globally, manufacturing ecosystems are facing the challenge of twin transition, i.e., how to utilize digitalization for improving or transforming the sustainability of manufacturing operations. Here, operations refer widely to the upstream of manufacturing, while the entire product lifecycle also covers the downstream and end-of-life operations. Here, sustainability is understood to consider the impact of the product lifecycle at environmental, social, and governance (ESG) levels. In this article, we explore this progress through the digitalization concept of business-to-business data sharing, and through one example of a manufacturing ecosystem in Finland. We discuss the federated data space concept and the international data spaces (IDS) architecture as technological building blocks of twin transition, and report the first results from an industry−research shared-risk project. Semi-structured interviews and a diary-style reporting from an industry−research IDS proof-of-concept (PoC) experiment are presented and analyzed within a design science research method framework. The findings give the first indications that while data sharing is seen as important and increasing in relevance in industry, it is currently challenging for companies to see how an open standard architecture creates value beyond a single limited ecosystem view. We also highlight possible avenues for further research.

## 1. Introduction

For the next decade, the manufacturing industry will be faced with the challenge of twin transition [1]. This twin transition is translated as the need to increasingly tackle sustainability challenges and to change corporate culture towards sustainable principles, while utilizing digitalization—understood here as highly malleable compositions of established key enabling technologies such as IoT, cloud, AI, and data management in general—in the process. This is especially evident in the case of CO_2_ emissions reporting and validation from the entire value chain of manufacturing, where approaches such as Scope 1, 2, and 3 are crucial for companies for verifying their emission reduction targets in practice (https://ghgprotocol.org/, accessed on 11 April 2023).

The organization of manufacturing companies and their key partners into different types of ecosystems has been on a swift rise during the last decade [2,3]. Three main types of ecosystems are identified: Business ecosystems organized around a central firm, with business activities tying other partners to the business operations. Innovation ecosystems centered around an innovation or value proposition, without the need for a clear central firm. Platform ecosystems, where firms are organized around a specific technology platform. In this sense, investigations regarding the twin transition, including its prerequisites, implementation, and impact, are feasible to be carried out at the level of the ecosystem.

Sustainability as a concept encompasses environmental, economic, and social dimensions, all of which can also have interdependencies [4]. Because of this, sustainability challenges are labeled as wicked problems, i.e., those that no single company can solve alone, and which are solved gradually over time, in multiple phases. Thus, they necessitate investigation at an ecosystem level. Recent research has emphasized that a reflective ecosystem practice is crucial for maintaining a continuous balance between innovation and business ecosystems of a certain group of companies and actors [5].

In the process of twin transition, data and data economy are seen as crucial enablers for increasing transparency and traceability, for example in the form of digital product passports [6]. Within the EU alone, the business value of data economy is expected to be around 800 billion EUR by the year 2025, and the business-to-business value of data is a significant part of this amount (https://commission.europa.eu/strategy-and-policy/priorities-2019-2024/europe-fit-digital-age/european-data-strategy_en, accessed on 11 April 2023). As a result of technological progress and willingness to stay competitive in markets, manufacturing companies have implemented data collection architectures within the governmental innovation frameworks of Industry 4.0 (Europe), Made in USA (USA), or Made in China 2025 (China). This results in manufacturing companies having cumulative storage of data, for which the utilization towards different business purposes is now gaining traction. Figure 1 illustrates how sustainable and circular manufacturing requires data sharing as a key enabler [7], and how value from data is born through trusted collaboration [8].

One technological concept that is gaining momentum within EU member countries is a federated data space [9]. It is directly based on the principle of data sovereignty, meaning that the owner of a data asset always stays in control regarding who gains access to said asset or how it is being used. Data sovereignty on the other hand is a key principle of the EU Data Strategy, which defines how the EU as a single market area wants to utilize data in sustainability and business objectives in the future [10]. Considering ecosystem research, federated data space defines ecosystems as networks that share data with common principles in order to reach jointly agreed upon objectives. In this paper, we focus on an open-source federated data space architecture and implementation called the International Data Spaces (IDS), which is coordinated by the International Data Spaces Association (IDSA) (https://internationaldataspaces.org/, accessed on 11 April 2023).

Several research projects and government initiatives have started developing federated data spaces for various fields. This has already resulted in several use cases from different starting points and with different purposes. One well-known solution is a data space for sharing data in the supply chain, called the Smart Connected Supplier Network (SCSN) (https://smart-connected.nl/en, accessed on 11 April 2023). SCSN is a data-sharing ecosystem that supports the manufacturing value chain, where participants are interconnected. In SCSN ecosystems, different actors, namely manufacturing companies and service providers, operate according to a common platform and regulations in order to enable data sharing. Service providers have a central role in enacting a data space by providing manufacturing companies with IDS connectors, whereby companies could exchange data [11]. In the mobility sector, the Mobility Data Space (MDS) (https://mobility-dataspace.eu/, accessed on 11 April 2023) brings together participants who want to monetize their data or need data for innovative mobility solutions [12]. MDS connects actors who provide mobility data with other actors who need data to develop and form new business models with full security and information sovereignty. Catena-X is an initiative that provides an environment for the creation, operation, and collaborative use of end-to-end data chains along the entire automotive value chain (https://catena-x.net/en/about-us, accessed on 11 April 2023).

In this paper, we explore how the concept of a federated data space, built on the EU values of open standards and data sovereignty, is perceived as a value-adding component in manufacturing ecosystems that are pursuing solutions for sustainability challenges. Here, we focus on the upstream operations (supply chain and manufacturing) of a Finnish manufacturing ecosystem as an example. At the same time, we fully recognize that the motivation for B2B data sharing stems from the need to examine the entire lifecycle of manufactured products (product lifecycle management (PLM)), including both upstream, downstream, and end-of-life [7]. We first look at the motivation and theoretical background of federated data spaces. Next, we utilize the International Data Spaces reference architecture model (IDS-RAM) to illustrate how the federated data space concept turns into an implementable set of processes and technical building blocks. Finally, we use a Finnish manufacturing ecosystem as an example of an empirical case to investigate how the perception of value is formed. Investigation is done through a collection of semi-structured interviews with different partners of the ecosystem, as well as through a research diary on the deployment of IDS software components to selected ecosystem companies. The ecosystem in question explores federated data spaces in the context of a jointly funded innovation project, where public sector funding balances the risk and allows for experimentation work with concepts and tools that have a lower maturity level. This article should be understood as a status report from an ongoing industry−research collaboration project, where additional results are expected to be published in later articles.

## 2. Materials and Methods

### 2.1. Research Objectives and Questions

Engaging federated data spaces to make a sustainability impact and capture business value is a strategic choice for manufacturing ecosystems and requires that multiple phases and viewpoints be covered. On a strategic level, the management levels of each ecosystem partner company need to understand federated data spaces as both a functional principle as well as a business opportunity before any investment decision can take place. In the beginning of an investment phase, there needs to be an onboarding plan for each ecosystem company to handle the practical implementation of the federated data space concept to business processes. This onboarding consists, for example, of CAPEX/OPEX investments, upskilling the workforce and ensuring trust and interoperability between participating companies in terms of the security and semantics of data. Within this whole onboarding process, companies of different sizes need to be kept onboard, for example, by offering support processes for SME-sized players, to ensure that all ecosystem companies can participate in value creation and capture in the federated data space.

Our research objective is to better understand this process of engaging and utilizing federated data spaces from a manufacturing ecosystem point of view. Research is motivated by the potential of data sharing for both increasing business and staying competitive, as well as for implementing the sustainability measures imposed by regulators. We adopt a design science research method (DSRM) where we work closely together with companies, map the progress of their understanding of data sharing, and conduct joint piloting activities to especially understand the technological onboarding process better. As a key outcome, we want to uncover the practical ways in which data sharing and federated data spaces create and capture value for manufacturing ecosystems.

Our main research questions (RQ) are as follows:

RQ1. What is the strategic relevance of data sharing for sustainable manufacturing ecosystems?

RQ2. What is the perception of a federated data space concept from a sustainable manufacturing ecosystem point of view?

RQ3. What are the main challenges that manufacturing ecosystems are facing in the adoption of a federated data space concept?

We are seeking answers to these questions through our own position as a national innovation hub for IDS and the resulting research and piloting activities, as well as through an example manufacturing ecosystem in Finland, conducting jointly funded shared risk experimentation with IDS.

### 2.2. Federated Data Spaces

Digital transformation is creating a data ecosystem with data regarding every aspect of our world, spread across a range of intelligent systems, with structured and unstructured data that can be exploited by data-driven intelligent systems to deliver value [13]. For example, one of the main challenges presented for Industry 4.0 is the representation and seamless exchange of data originating from heterogeneous elements, often from very different, albeit related, action levels [14]. Therefore, there is a need to bring together data from multiple sources and to support knowledge sharing among participants within data ecosystems [15]. Within this data ecosystem concept, the usual way of organizing the business has been for a single large platform company to seize the market and almost exclusively capture the resulting value through tight vertical integration. In the EU, the recognition of this development has led to a new initiative called the EU Data Strategy [10] to balance the market and open the value capturing possibilities to a wider group of players of different sizes and specializations. A key principle in the strategy is data sovereignty, which states that the owner of data should always stay in control of who is utilizing the data and how. This strategy is especially aimed towards business-to-business data sharing, of which manufacturing ecosystems are one example.

The concept of a federated data space has been gaining traction to address the challenge of limited market access in a way that conforms with the EU Data Strategy principles. A federated data space can be defined as a decentralized infrastructure for trustworthy data sharing and exchange in data ecosystems based in commonly agreed principles [16]. Furthermore, federated data spaces can be viewed as an ecosystem of data models, datasets, ontologies, data sharing contracts, and specialized data management services together with soft competencies including governance, social interactions, and business processes [17]. Certain principles are required for each federated data space [9]:No middleman—no powerful (commercial) parties that determine only what the rules of the data space are.Decentralization—data are not stored centrally, but at the source and are thus only shared (via semantic interoperability) when necessary.Data sovereignty—the data owner can control the use of their data and capture value from it.Neutrality—different actors and stakeholders are treated equally within a dataspace.Identity management—refers to the identification, authentication, and authorization of organizations or persons.Traceability—transactions made within the distributed system should be recorded and logged.Data interoperability—the ability to establish a mutual understanding on the structure and semantic meaning of data.Data findability—the ability to discover available data services through, e.g., service catalogues.

The International Data Space (IDS) standardization initiative aims at addressing the above-listed requirements. It was established in 2014, and currently has a network of 130+ members representing both industry and research. They actively develop an architecture reference model, for which partner members can make software component implementations on top.

### 2.3. IDS Architecture

International data spaces (IDS) enable the sovereign and self-determined exchange of data via a standardized connection across company boundaries. It provides a foundation for facilitating innovative cross-company business processes and creating smart-service scenarios, and guaranteeing data sovereignty for the data owner [18]. In general, IDS aims at addressing the many challenges in the overarching use of data in terms of interoperability, transparency, trust, security, and adaptation by a critical mass [19].

One of the key principles of IDS is that data remains with the data owner until it is needed by a trusted business partner and when the data are shared, terms of use can be linked to the data [20]. IDS provides all participants with a common ecosystem that allows and supports the establishment and maintenance of each participant’s digital sovereignty [21]. This means that IDS participants can both share data with business partners in an interoperable way, as well as retain self-determination with regards to these data assets [22].

IDS defines a technology-agnostic architecture, which is described and continuously updated in the IDSA Reference Architecture Model (RAM) (https://internationaldataspaces.org/publications/ids-ram/, accessed on 11 April 2023). IDS-RAM defines the required standards, control, and enforcement rules for data exchange among different participants in a data space, specifying their components and mechanisms [22]. In Figure 2, the most important modules and components of the IDS reference architecture model are depicted [23].

As described above, the IDS reference architecture aims at defining the necessary components needed for establishing data spaces. In the following section, the most important modules of the IDS architecture are explained in more detail.

Connector—An IDS-based data space consists of connectors via which data providers and data consumers are connected to the data spaces. Data are always exchanged in a point-to-point manner, from one connector to another. With connectors, every data provider can also define the rules and conditions (usage policies) under which data are shared with a data consumer. These rules include scenarios, e.g., restriction of data usage for a specific group of participants, restriction of usage to specific purposes, or the usage of data not more than N times. After a data consumer requests a data set from the provider, a contract negotiation process is started, during which the usage policies are negotiated. A successful contract negotiation leads to a contract agreement, which entitles the data consumer to access the data resource offered by the data provider.

Identity provider—The identity provider consists of three components, namely the certificate authority (CA), the dynamic attributes provisioning service (DAPS), and the participant information service (ParIS). The role of the CA is to grant X.509 (an International Telecommunication Union (ITU) standard) certificates, i.e., machine-readable certificates that safeguard against man-in-the-middle attacks. DAPS enables connectors to authenticate themselves using those X.509 certificates received from the CA. After successful authentication, the DAPS issues OAuth2 (“Open Authorization”, an open standard for access delegation) access tokens for connectors, who need these tokens to access the services and data of other connectors on the user’s behalf. ParIS components hold general information of all participants of a data space.

Metadata Broker—The broker component stores information about the data endpoints offered by participants of the data space. It does not store the data sets itself, but only the meta information and is thus sometimes referred as the metadata broker. The broker provides a query interface for a connector and can hence be considered as the search engine of data space information. This means that data providers can register their data sets at the broker and, subsequently, data consumers can search for data they need for their use for case-specific purposes. It should also be noted that the broker is an optional component within a data space because the connection between two participants also can be established directly, in case they are known to each other.

App Store—The App Store is a component where participants can search and discover applications for data transformations that are deployable in their connectors. Participants of the data space can make standard applications visible in the app store. Any other participant interested in this functionality can discover the app and deploy it in the connector.

Clearing House—The clearing house is a service for logging data exchange transactions within IDS. Clearing house is a cross-domain service that receives information about transactions, participants, and references to any existing legal contracts; stores this information in a verifiable form; and makes it available to participants. The clearing house leads to a higher trustworthiness as data flow can be proofed by a neutral third party. It also can be used to establish pay-per-use or, better said, “pay-per-transfer” business models.

Vocabulary Provider—In order to better define one’s own data, domain-specific vocabularies can be created and made available through the vocabulary provider component. It manages and offers vocabularies that can be used to annotate and describe datasets. In particular, the vocabulary provider provides the information model of IDS, as well as other domain-specific vocabularies.

Certification—Data security and data sovereignty are the fundamental characteristics of IDS. Participants within IDS must use certified software (e.g., the IDS Connector) to securely exchange data in a sovereign way. Furthermore, data can only be exchanged in an IDS-based data space, if the exchange takes place between trustworthy and certified participants. Therefore, the certification framework in IDS comprises two phases: certification of participants and core component certification.

### 2.4. IDS Information Model

The purpose of the information model is to provide the domain-agnostic common language for the international data spaces for automated or semiautomated exchange and sharing of resources within the ecosystem while preserving the data sovereignty of data owners. The information model comprises the first level of data interoperability between an ecosystem of companies. Second level is a domain and possibly ecosystem-specific data model that captures the key concepts and terms the ecosystem needs in their business processes. Ecosystems utilizing IDS need to implement both these levels to become fully operational in the federated data space.

A resource has a unique ID and can be traded and exchanged between remote participants using the IDS infrastructure. Examples of resources include documents, time series, messages, images, and media streams. Figure 3 illustrates a concern hexagon (C-hexagon), meaning a separation of concerns (SoC) listed as dimensions where each dimension can be considered independently [23].

Content—the most essential aspect, deals with: (i) resource description, (ii) representation in a machine-interpretable format, and (iii) materialization as instances.

Concept—covers meaning, annotation, and interpretation, e.g., (i) keywords of natural language, (ii) terms such as vocabularies, and (iii) types defined in ontologies or type systems.

Context—deals with (i) time, (ii) place (space), and (iii) real-world entities.

Communication—deals with communications (i) by sending messages, (ii) to a resource or service endpoint or an IDS connector, and (iii) to perform an operation.

Commodity—address the value and utility in terms of (i) provenance, (ii) quality, and (iii) policies.

Community of trust—refers to certified participants operating certified components such as connectors so that data exchange and sharing is done in a secure and trusted way in accordance with contracts composed of usage policies, thus ensuring data sovereignty.

### 2.5. Research Methodology

Our empirical study follows a design science approach and, more specifically, a six-phase process for understanding the research questions posited earlier [24]. Design science is primarily a problem-solving paradigm that addresses research through the building and evaluation of artifacts designed to meet identified real-world business needs, considering relevance and rigor. Rigor is achieved by appropriately applying existing scientific foundations and methodologies [25]. Furthermore, design science can be considered an especially suitable research paradigm for so-called “wicked problems” [26]. In this context, we consider the achievement of sustainability objectives as a wicked problem, and federated data spaces as one technological concept whose role needs to be understood better in this problem context. A federated data space is a complex mixture of design and technology that evolves over time, requires high levels of creativity and continuous innovation, and is characterized by unstable requirements and complex interactions among components.

The design science process is illustrated in Figure 4. It starts with problem identification and motivation, which in our context translates to a better understanding of the value of federated data spaces for manufacturing ecosystems, motivated by both future competitiveness as well as future (sustainability) regulation. We approach this by mapping the perceived importance of data sharing from our example ecosystem through semi-structured interviews. The second step is defining the objectives of a solution. In our context, this means surveying related research on the importance of data sharing, familiarizing with the IDS architecture model, and designing a series of joint proof-of-concept (PoC) experiments with the ecosystem companies. Third step is the design and development of an artifact that aims to solve the identified problem, and this translates to the actual development of PoCs. The demonstration phase in our context translates into joint experimentation of the PoCs against the business needs of the ecosystem companies. Evaluation consists of the feedback received from the companies from the experimentation phase, as well as our own reflections from the experiments. Communication takes place within the project consortium as well as through selected public channels, to disseminate the findings from the experiments. Finally, the findings are reflected to the original objectives and design for iterative improvements.

Design science aims to maintain the business relevance of identified problems through close industry−research collaboration. However, close collaboration is not self-evident as there may also be problems in cooperation or issues that hinder the practical collaboration. The authors of [27] identified, based on a literature review, the root causes of the low relevance and utility of research. They stated that the main causes are (1) researchers having simplistic views (or wrong assumptions) about SW engineering in practice, (2) lack of connection with industry, and (3) wrong identification of research problems. To avoid this, Garousi et al. stated that conducting industry-relevant research requires close industry−academia cooperation, all the way from the problem identification to delivering and publishing the results [28].

### 2.6. Target Ecosystem

Manufacturing supply chains are operating in a turbulent environment where delivery times are decreasing, quality requirements are becoming tighter, and resiliency is needed. Companies must be more open and comply with environmental requirements and develop their sustainability. On the other hand, digitalization brings a significant potential for companies to modernize processes, improve communication, and increase profitability. To address these aspects, the Open Smart Manufacturing Ecosystem (OSME) is a collaborative initiative that engages manufacturing companies to speed up the needed transformation by engaging, supporting, and leveraging the skills and strengths of its partners, as well as encouraging towards more open collaboration (https://www.mexfinland.org/osme/, accessed on 11 April 2023). The OSME project is scheduled for 2022 and 2023, and it consists of eight founding partners and six associate partners. One of the eight founding partners is a research organization, while the others are companies.

### 2.7. Semi-Structured Interviews

To obtain the first insights into the industry viewpoints and the current state, challenges, and vision regarding data sharing and re-use, five (5) semi-structured interviews were conducted between November 2022 and February 2023, with the interviewees drawn from the project industry partners. The interviewees were selected for their involvement in the data sharing and re-use activities in their respective organizations. The interviewees were grouped into three main categories based on their role in the data sharing ecosystem: principal, i.e., the manufacturing company that had the leading role in the ecosystem; suppliers, which are part of the supply chain for the principal; and technology providers, i.e., companies that would enable data sharing operation in a technical sense, as solution and/or service providers. Both the principal and the suppliers act as providers and users of shared data, and the technology providers develop solutions that facilitate data sharing. The list of interviewees is presented in Table 1.

The selected empirical method for the problem identification and motivation was a semi-structured interview, where the interviewer presented a topic for discussion and prepared it with a short descriptive statement and an open lead-in question, which the interviewee then explored as they want. For each topic, a set of additional exploratory questions was prepared to guide the exploration to cover essential subtopics, should the interviewee not address them on their own. The key themes to be covered in the interviews were drawn from challenges identified in the literature and related EU reports [17,29,30,31,32], as follows:Theme 1 (warm-up): Do you already currently share or re-use data with a partner?Theme 2: What opportunities, needs, or pressure do you anticipate regarding engaging in data sharing/re-use now or in the future?
○Elaboration topics: Strategic versus marginal, regular/systematic versus ad hoc, point-to-point versus multi-partner, and small versus large data volumes.Theme 3: How should data sharing/re-use be organized in your current business ecosystem?
○Elaboration topics: Data ownership, data sharing governance, security aspects, and partner types/roles.○Follow-up question: How well are these elements already in place in your current business ecosystem?Theme 4: What is the level of readiness and SWOT for data sharing and re-use?
○Elaboration topics: Data, people, technology, processes, your company, and current partners.Theme 5: SWOT regarding business risks, investments, trust, and culture (your company and partners)?Wrap-up: Was there anything we did not cover?

Interviewees were provided with a short overview of the interview session about a week in advance, containing a consent statement and the high-level themes to be discussed in the session. The consent, including interview recording, lawful purpose for processing, and the use and retention of data, was requested verbally in the interview and later in writing with the transcript. Each interview session lasted about 1 h and was conducted online using Microsoft TEAMS, with the session recorded using the tools built in to TEAMS. The interviews were conducted in Finnish. Two interviewers were present in 80% of sessions, one as the main interviewer guiding the discussion, the other as a back-up, to manage recording, take notes, and ensure that nothing essential was accidentally missed in the interview. Two interviewers also provided a level of technical resilience, to ensure that the interview could continue and to prevent the recording from stopping and being possibly lost in case an interviewer lost connection during the session. The session recording was transcribed afterwards by one of the interviewers, and the transcription was provided to the interviewee for comments and approval and for signing the consent. After the approval, the recording was deleted. The transcripts were pseudonymized prior to the analysis to mask data that could reveal the identity of interviewee or their organization.

The interview material analysis was carried out as a thematic analysis, with themes drawn from the interview topics defined prior to the interviews. The themes selected for the analysis were chosen on the grounds that they reflect the (1) current and future state of data sharing in the industry, and (2) how well the industry opinions aligned with the values EU were promoting for data economy. The specific themes were (1.1) the importance and extent of data sharing, (2.1) role of sustainability as a motivator for data sharing, (2.2) preferences to open standards versus closed ecosystems, and (2.3) opinions regarding data sovereignty.

The interview transcripts were read through to find statements where the interviewee discussed the relevant theme and the statement analyzed to pick up the aspects the interviewee brought up regarding the theme. For the most part, the statements were directly about the theme, but in some cases, they were indirect/oblique and required interpretation, e.g., an interviewee from a technology company stated that they were investing in the unification of APIs, which implies that they predicted an increased importance and volume in data sharing in their customer base. The identified aspects were then summarized to obtain a set of viewpoints to the themes, as well as grouped per role of the interviewed company (principal, supplier, and technology provider) to see if there were role-dependent differences in the viewpoints. Quotations illustrating the viewpoints were selected and translated to English for this article.

### 2.8. Problem Identification and Motivation

In our focus ecosystem, the process of first IDS PoC started in the beginning of October 2022 and the evaluation phase ended in February 2023. IDS PoC development was organized as one joint team consisting of research and industrial representatives. The joint team had a status meeting bi-weekly and practical working meetings whenever needed. Furthermore, Slack and Jira were used to plan, communicate, and discuss practical development issues.

The DSRM process starts with the definition of the specific research problem and the justification of the value of a solution [24]. The first level of problem/value setting is the coming together of an industry/research consortium in seeking risk funding to investigate the data-sharing phenomenon in more detail. The second level is the collection of interview data, where industry partners analyzed the data sharing status, future impact, and associated risks/challenges/obstacles. Based on this background, the OSME project defined an overall industrial challenge for data sharing: how to securely exchange the manufacturing information needed in an open manufacturing ecosystem. The focal company of the manufacturing ecosystem (OEM) acted as the owner of the challenge and encouraged different actors to cooperate to solve the challenge. As one part of the challenge, empirical data sharing proof-of-concepts (PoC) were designed, implemented, and evaluated within the consortium.

However, based on the discussions/interviews with the focal company, it become evident that the value promise of open standard-based data sharing architectures is unclear. What does their implementation require from an industrial company? The initial problem framing was discussed in the PoC start-up meeting with the representatives of the focal company and research partner. In the current setting, OEM uses Excel to send information about the needs of the different parts to the engine part suppliers. The need for parts is either related to the orders OEM has already received or forecasts of future orders (e.g., OEM leads). With the help of Excel, suppliers can plan and forecast, for example, their own production capacity in the future. The supplier will search/filter related data fields from Excel—and this is time consuming and error prone.

To tackle this, OEM built a solution where suppliers access the information through an API. OEM’s intention is to have one interface for all suppliers and not to do case-by-case customization. This is sufficient for now, but suffers from low scalability. In the future, there may be a need to limit what information different suppliers receive. If, for example, 150 suppliers are connected to the interface and competitors are involved, there may be a risk that the suppliers will receive sensitive data in an unsolicited manner through the interface. For these reasons, data sovereignty, access, and security must be handled in a relevant manner.

Outcome: The research problems identified and formulated during the problem identification and the motivation phase were as follows:How to securely exchange manufacturing information needed in the manufacturing ecosystem?How does the implementation of open standard-based data sharing architecture take place in an industrial context in practice?

### 2.9. Define Objectives of the Solution

The objectives should be inferred rationally from the problem specification [24]. IDS architecture was chosen for open standard-based data sharing architecture because its components are open-source code, it is based on the EU Data Strategy, and its maturity level is currently sufficient for piloting. The objective is to implement data sharing architecture (IDS) into the industrial environment. Furthermore, the objective is to gather experiences from the industrial implementation process. As IDS is still in its early phase [17], it is evident that we need to gather and distribute experiences from IDS implementations and build awareness of the benefits of the IDS vision [33]. IDS brings the data governance layer between the focal company and suppliers. The target of the IDS connector was to filter the information that different suppliers received from the focal company. It is also necessary to define and test how the new supplier is added to the connector (technical and possible user interface solution). It was anticipated that IDS brings data governance and more modularity to the focal company’s overall solution, as well as helps to manage API modification needs.

Outcome: The identified objectives of the solution included the following:Implement open standard-based data sharing architecture (IDS) in an industrial context.Gather experiences from practical IDS implementation processes.

### 2.10. Design and Development

Based on the identified objectives for a solution, a proof-of-concept prototype for an IDS data sharing solution was designed and developed in an agile manner. The practical implementation and demonstration were done by a team of five developers (two from the focal company and three from the research lab). Tasks were defined and managed in Jira and issues were discussed in Slack. The architecture of the solution is described in Figure 5.

As can be seen in Figure 5, in the PoC, the focal company acts as a “data provider” and can be considered as an OEM in the manufacturing ecosystem. OEM has its own IDS connector that retrieves data from relevant management systems, including SAP and ERP, for example. The integration between the management systems and the IDS connector is implemented with a REST API that enables submitting data requests to management systems. Furthermore, the OEM’s IDS connector contains specific data filters that determine the exact data items that can be accessed by different suppliers. In this way, OEM can specifically control the data that are visible for each supplier. In this case, the filters can be considered as pre-specified requests to the API, which include supplier-specific parameters.

To be able to receive data from the data provider (i.e., the OEM), the suppliers are required to deploy their own IDS connectors. As previously discussed, all data exchange within IDS-based data spaces occur between connectors that need valid certificates that ensure their compatibility with the IDS specifications. Before the supplier can access data resources defined by the data provider, they need to establish a contract agreement with the provider to receive a reference for the data. Once the contract agreement is made, suppliers can repeatedly access the data using the same reference link. If the data are updated by the data provider, suppliers are always provided with the most recent version of the data. Suppliers can also subscribe themselves as listeners for data changes. In this way, a supplier is automatically notified whenever the data relevant for that supplier is being updated by the data provider.

The top part of Figure 5 depicts the support services provided by IDS. These include, for example, the identity provider and the metadata broker. These services are utilized by both data providers and data consumers, and they aim at ensuring the correct functionality and trustfulness of the data space. Moreover, they support the scalability of the data space by ensuring the identity of new data exchange participants and allow for both storing and searching the metadata regarding the available data resources.

Outcome:

The artefact designed and developed was IDS solution for providing secure data sharing and supplier-specific filtering. The technical system implementation of the solution included the following:Data provider IDS connectorData Receiver IDS connector (two connectors)IDS support services

### 2.11. Demonstration

Demonstration uses the solution to solve one or many instances of the problem [24]. The implementation and installation of the PoC solution was demonstrated and evaluated using realistic industrial context (focal company/OEM). The demonstrator encompassed three IDS connectors and required IDS support services that established a minimum viable data space. Next, the demonstration configuration is explained in more detail.

IDS connector 1 was installed to the OEM’s IT infrastructure (Figure 5). The connector was deployed in a Kubernetes cluster and a needed communication endpoint was registered and opened to enable access to the connector from the other connectors. Additionally, two data resources were created for the connector. These data resources were integrated with the OEM’s API providing data about the production program of the OEM. For both data resources, separate data extraction parameters were specified that enabled differentiating the exact data that are offered for different data consumers (i.e., suppliers). Finally, a dedicated certification was created for the connector.

IDS connectors 2 and 3 were installed to a server infrastructure deployed on a research lab. These connectors simulated two different suppliers that should receive production data relevant for their business environment. Separate certifications were created for the connectors, as well as unique IDs. With these IDs, the data provider was able to allow define access policies between different data resources and different suppliers. Finally, on the servers hosting the connectors, required communication endpoints were created and opened to allow for data traffic between the connectors.

IDS identity provider and metadata broker were installed and deployed on a third server instance at the research lab. The metadata broker enabled OEM to discover information about available suppliers as well as suppliers to retrieve information about the data resources offered by the OEM. Furthermore, on the identity provider, the connectors were able to authenticate themselves with their X.509v3 certificates and subsequently receive an access token in exchange for which they needed to access the services and data of other connectors.

As earlier discussed, one of the main objectives of the demonstration was to test and validate data exchange between the data provider’s (i.e., the OEM) and data consumer’s (i.e., the suppliers) IDS connectors. Furthermore, based on the identity of a supplier, different data sets were supposed to be exchanged between the connectors. This kind of identity-based data filtering was implemented by creating a dedicated data resource for each of the suppliers into OEM’s IDS connector. The data resources were configured with different parameters that define the exact data that were retrieved from OEM’s API. This approach has two benefits:OEM can selectively disclose data from their production environment to different suppliers without the fear of data leakages.The suppliers receives only that data that are essential for them and there is no need to locally extract meaningful insights and valuable information from big data sets.

In the demonstration, it is shown how data resources with different data extraction parameters can be created for different suppliers on the OEM’s side and how suppliers can retrieve data relevant for them from OEM’s IDS connector.

Outcome: The PoC artefacts designed and developed were installed into an industrial environment consisting of an industrial focal company (OEM), a supplier (imitated by a research lab), and a broker (a research lab).

### 2.12. Evaluation

In the evaluation phase, it was observed and measured how well the artifact supported a solution to the problem [24]. The developers recorded in their research diary what they had done and all their observations and comments during the development and installation work. This included step-by-step technical implementation description as well as observations. PoC successfully demonstrated IDS implementation in an industrial context and increased the focal company’s understanding of IDS adoption and applicability. The detailed PoC experiences are described in Section 3.2.

Outcome: See Section 3.2.

### 2.13. Communication

The industrial problem and its importance, the artifact, its utility and novelty, the rigor of its design, and its effectiveness will be communicated to the research and professional audience [24]. The status and results of the IDS PoC work were communicated via presentations throughout the process to the OSME innovation ecosystem project. The results of PoC will be communicated in scientific community as well as in professional communities such as IDSA and its national representation IDSA-Finland (https://www.idsa-finland.fi/, accessed on 11 April 2023).

Outcome: Professional communication in IDSA, IDSA-Finland, and OSME innovation ecosystem project. Scientific communication in appropriate data spaces forums.

## 3. Results

### 3.1. Interviews

The purpose of the interviews was to gain insight into the industry mindset and viewpoints on data sharing and re-use, in particular the importance and use of data, how sharing and re-use is seen to evolve going forward, and how the industry mindset aligns with that of the EU, using three topics (sustainability, open standards, and data sovereignty) as specific probes.

All interviewees saw that data sharing and re-use has, and will have, increasing importance for their business. However, there were differences in how strategic it was seen to be at the moment. The representatives of the principal company saw that the data sharing and re-use alone were not strategic, but as a means to achieve strategic goals. “It’s not like [data] is its own agenda […] we don’t state that data should be shared, but that we should use digitalization, where it makes sense” [I.1] (I.1 is the interviewee ID, see Table 1). “But as a means it has a growing importance—to the extent that the company wants to be on the forefront on this matter. [The company] operates in such sectors that … evolution and understanding of regulations is of primary interest, so [the company] … wants to be … in the fore rather than a follower” [I.4]. Data re-use already had a significant role in their business, and the role of data sharing was increasing. However, the viewpoint of data sharing was that while the volume and frequency would increase, there was a desire to limit the volume, and this would lead to sharing information rather than the raw data. “It will increase … it will be on a daily basis … there will be a lot of sharing … but it is also up to … sharing information or insight, or just data ... it’s not like it’s always going to increase, there’s also the desire to push it down, the volume” [I.1]. “Once you can efficiently crystallize and turn it into information, it will have more added value than that mass of data on its own, so there’s the benefit” [I.4]. The principal company also brought up that dependency on external data can and has already become so central to some processes and services that they cannot survive without it. “If for some reason we wouldn’t get [the data] then in practice this service that we’re providing, it wouldn’t simply work as it should … so if you don’t get the data … it can be … quite fatal to the company and its processes” [I.4].

The technology providers saw the importance primarily through their customer’s needs and the potential it represented for their own business. “It is very strategic … we see that data is very fragmented on the customer side. We hear that they need end-to-end systems … so we see the upsell potential here” [I.2]. “In the end [the strategic importance] comes from our products and cloud services. Technical integrations have been done for a long time, but its importance seems to be continuously increasing in our customer base and through that this area has a growing importance for us” [I.3]. There was a clear belief that the market was growing and would be big. “In 2023 we have a big market here already [I.2]”. At some point it is self-evident that data is essential part of the business and then the question is … the relationship between our physical products and data” [I.3].

The supplier commented that the data do not yet have a strategic role: “[Data sharing] is really in its infancy still. [Data re-use] is not … central part [of the strategy] yet” [I.5]. However, the importance and amount of sharing would increase. “It will be emphasized, definitely. Data sharing will increase and it will be used to make business too, it’s obvious as I see it” [I.5]. Yet it would not grow indefinitely as it depends on whether or not there is a business case: “There is already a stupendous amount of data available … it’s not all being used and it’s not understood and people don’t know how to use it, but it doesn’t necessarily have business in it either” [I.5].

Uses for the data brought up by the interviewees included:Better understanding, transparency, and situational awareness of the operation and value chain [I.1; I.2; I.4; I.5],Enabling different parties in a supply chain to solve customer issues [I.1; I.3],Bringing an ability to respond to external requests and requirements [I.1; I.3; I.5],Reach sustainability goals and use data as proof of that [I.1; I.3; I.5],Achieve resilience and agility to react to the changes in the market and geopolitical situation [I.1; I.2; I.4],Flexibility, including the trend towards high-mix low-volume production [I.2; I.5],Bring business growth through new added-value services, operational excellence, and quality improvements [I.2; I.4],Support employees in their daily work and through that increase the meaningfulness and engagement of the work [I.2].

In general, this all is seen through a business lens: “And all this of course leads to business performance. That is ... that’s how the thinking goes” [I.2].

We wanted to also probe the viewpoints to three EU value-related topics—sustainability of operations, open standards for data sharing, and data sovereignty—to see how well the industry thoughts aligned with the EU-level vision. Sustainability was addressed through the EU twin transition, open standard adhered to the EU Data Strategy vision of cross-domain interoperability (a topic also for future research), and data sovereignty addressed the immutable rights of the data owner to the way value is created from said data.

All of the interviewees recognized that sustainability is a factor motivating data sharing and there were some indications that it could be considered a direct motivator as well. For instance: “All these regulations that are related to sustainability … is there something like larger megatrends behind here clearly. And this will … increase the people’s overall awareness of things” [I.2]. Two interviewees also brought up the societal viewpoints: “There’s also the societal importance of it there too” [I.4]. “In the future such [sustainability-related] requirements will come from customers and … from the society, and that you can show that you as an operator act sustainably [has an impact]” [I.5]. Its role as a primary motivator can, however, be questioned: “Not sure though whether [the motivation to share data] comes from the environmental sustainability requirements and regulation side, or does it come more from the [business] value side” [I.1]. In general, the interviewees viewed sustainability topics through business-related motivations. These topics can rise as requirements from customers or other stakeholders, such as financing institutions: “[EcoVadis, a 3rd party] … rates companies for their sustainability and [this rating] is used by financiers and others [I.5]”. Sustainability can also be question of competitiveness: “The carbon footprint of the entire supply chain and production … is highlighted in the regulations and it is good for us on the other hand. When you’re mature on that area, it provides competitive advantage” [I.4].

The EU preference is towards open standards that facilitate cross-domain data sharing across the business landscape. The interviewee opinions on this were more varied. The Supplier representative was clearly calling for open standards, as they deal with multiple principals and having to accommodate, e.g., different API’s for each of them separately is a complication: “We have 3–4 different … international big companies and all of them have their own systems … easy interface to the systems they offer … if there would be … some sort of standard then it would make it easier.” [I.5]. Likewise, representative of one of the Technology providers brought forth similar thoughts: “It is easier for the customer … there could be these kind of platforms around the data, open interfaces … to make it possible, data structures … well understood and well defined” [I.2]. From the Principal side the stand was less clear, the focus was more on individual ecosystem and whether the sector (e.g., manufacturing business) is able to define its own data sharing systems or whether there will be an outsider who creates a disruption and everyone has to join that platform to perform data sharing: “The question is … is it someone else who comes to disrupt things … someone who offers [data sharing] as a service and in a way rules and in the end takes control over the network, or can the sector … go to that direction on its own without someone coming and making a full-scale disruption” [I.1]. But at least sectoral cross-sharing was envisioned, which can be seen as a nod towards in-sector open standards: “And then I see also one aspect that … there’s cross-sectoral … in-sectoral … manufacturing industry internal, data sharing between different companies in the network” [I.4]. The other Technology provider did not take direct stand on this but the statement on proprietary standards could indicate that they see closed ecosystem as a competitive strategy: “We have used different standards … so that we can have expanded those standards … but of course this requires that it’s our system at both ends” [I.3].

On data sovereignty, the opinions of the Supplier and the Technology providers were clearly approving, in the manner where the data originator is the data owner who has the full right to lend, lease, or sell the data to someone for further processing: “How I’ve always thought it is that who has originally created the data, they would own it—that’s how I understand the things should be” [I.2]. “The source is responsible for something and or owns something about it and from that onwards” [I.3]. “A clear guideline here is that the data originator is the data owner. I can’t see any other way for this. And that party then does whatever they want with the data, they can sell the rights to a 3rd party for their use, or for their development, or something else. But … the data originator owns it” [I.5]. Both Principal representatives acknowledged the need for control over data: “Governance must exist and … ownership” [I.1]. “It must not be [an open issue] to any actor how a given data set … can be handled” [I.5]. Reservations were also brought up with concerns that the ownership and regulations can hinder the ability to use the data: “[The governance] must allow … proactivity … in the process. And how to balance these is maybe [a challenge]” [I.1]. “Regulation is increasing around … data. How the data can be used, and who has the right to it. It is both [a threat and a positive thing]. It clarifies things, yes” [I.4].

### 3.2. Industry−Research PoC Experiment

The first OSME data sharing PoC was set up in a real OEM IT infrastructure in a cloud execution environment involving containerized applications with automated deployment, scaling, and management software engineering frameworks based on Docker and Kubernetes, respectively. As Figure 6 depicts, the data producing applications were concealed from other applications, and there was no direct app-to-app connectivity. Instead, applications were accessible via well-defined and controlled API management procedures based on the OpenAPI standard (https://spec.openapis.org/oas/latest.html, accessed on 11 April 2023).

Similar to any other SW application in the overall OEM system, the IDS connector in the data producer role was implemented as a containerized application compliant with the OEM IT infrastructure framework requirements. It interacted with data producer applications via an OpenAPI-based connectivity, and in turn published an IDS compliant interface to the data consumer applications in the Supplier IT infrastructure.

A lesson learnt was that identifying technical details in the containerized OpenAPI-based implementation with respect to the essential mandatory one-to-one counterparts in the IDS Connector adaptation, including, but not limited to, items such as inter-SW component authentication and authorization and actual data, needed to be analyzed carefully prior to any implementation actions. This is a vital initial cornerstone in deploying the IDS connector concept for data exchange successfully. On the other hand, this can become a stumbling block in identifying and spotting the immediate added value originating from the IDS concept, while the more extensive potential is in the re-use of the already implemented data sharing baseline in the long term.

## 4. Discussion

### 4.1. Value of Data Sharing and Open Standards

In general, we found that the industry representatives interviewed see the value of data, and the need for its movement across at least the immediate business ecosystem and possibly beyond. Although some actors may be more aligned with the vision EU for data sharing, and others can see downsides in that direction, we could not see clear conflicts with the EU values in the industry opinions. The implication of this is that the industry will likely seek to accommodate the EU viewpoint, even if it may require some business strategy shifts and letting go of some possibilities for data re-use.

There is a clear and unanimous view that data will become ever more central to business, and there is a growing need to share and re-use it effectively and efficiently both internally and externally. In some cases, the data are already a central part of the process or offering, to the degree that it cannot operate without the needed data. As is natural, the technology providers see the importance for it through the lens of market growth for data sharing and re-use solutions, whereas the principal and supplier are more focused on how the data are being used. Most of the uses identified revolved around operational excellence, but, interestingly, a technology provider also brought up that having access and visibility to data could also increase the meaningfulness and engagement of the work at hand. The supplier had some reservations regarding how widespread the data sharing would eventually be, although it is possible that the interviewee was thinking more about openly available data at this point, rather than data sharing within an ecosystem with a business purpose.

Sustainability did not seem to come through as a primary motivator. All interviewees acknowledged it as a factor, but it played its role mostly through external pressure, which was either a challenge or gave a competitive advantage if the company had solutions for this. However, this is a natural and expected viewpoint for a commercial operator—sustainability has a societal role, but money is what speaks in the end. On the two other EU value-related items there seemed to be a slight difference between the supplier, preferring open standards and strict ownership of data, versus the principal, who put less emphasis on openness and had reservations of data ownership versus data re-use. This could mean that the principal saw an upside in re-using the data from its supply chain, and would like to have a free hand on how they used it, whereas the supplier saw data as a valuable asset that they did not want to give away for free. The technology providers agreed with the supplier on ownership, but interestingly had two possibly differing stands on open standards, which could reflect the differences in their business offering and strategy.

### 4.2. Barriers, Limitations, and Risks

Perhaps the greatest individual risk perceived by the companies towards IDS was its maturity level, coupled with its long-term continuity. In a shared-risk environment, companies were eager to investigate and experiment with an architecture such as IDS and understood its value promise on a high level. However, the key questions are as follows: When is it applicable to implement IDS to production-grade environments? Will it perform in a stable enough manner? Will it have continuing support in terms of SW updates and other maintenance. Our ecosystem companies seemed to understand that the playing field of data sharing architectures has now been opened and there are no guarantees yet which candidates will emerge as winners, and what kind of consolidation steps will happen on the way.

A clear barrier perceived by the SME-sized players of the ecosystem is that they lacked in-house resourcing to implement and maintain an IT infrastructure required for interfacing an IDS-based federated data space. One option is to look for this implementation and maintenance as a service, which technology companies are interested to provide, but even then, the return on investment needs to be clear to justify the costs. One clear shock that would push for adoption is a focal company that adopts a federated data space model, and only agrees to exchange data with suppliers through it. Investment into it as a service IT infrastructure would be justified if the other option is losing a key customer.

One barrier is the upskilling of the workforce in the companies to successfully utilize the data and information acquired through the federated data spaces. In the interviews, companies listed transparency, production planning, and increased sustainability-related situation awareness as their business goals. Reaching these goals through data sharing requires personnel that can work with the data as part of their everyday operations. In informal discussions with companies, they saw this upskilling mostly as a demographic question, as the new and younger workforce is thought to be naturally more inclined towards working with data than the older generations.

Another barrier is related to the interoperability and ethical concerns of data sharing. Interoperability on the data sharing level requires common data and metadata models, and each ecosystem may need to develop tailored data models for their own use, representing a significant upfront investment. The IDS concern hexagon presented in Figure 4 is a domain-agnostic model that still requires domain and/or ecosystem-specific models on top of it. Ethical concerns can arise when data are uncontrollably leaked to other parties, either by accident or through malicious intent. In any data sharing system, these leaks need to be included as part of the overall risk management scheme.

Finally, the interview data used in this study were collected from a group of companies that already expressed an interest in data sharing and re-use and were collaborating in a nationally funded research project to that end. As such, it represents a view of those who are on the forefront of data sharing and re-use, and is limited to a national context and view, rather than a broad insight regarding the current state of data sharing and re-use in the European manufacturing sector in general. Furthermore, the findings can also correlate strongly between companies because they may stem from discussions between the project partners, and hence have already shaped the thinking and reflect a shared understanding of key concepts.

### 4.3. Utilization of IDS Functionality

Within the interviews and the PoC experiments so far, the companies of our focus ecosystem saw data sharing mostly through IDS connectors and the shared ontology and terminology. In other words, the elementary level of data sharing and data interoperability were addressed. Further functionalities in the IDS architecture should include, for example, the data marketplace, smart contracts for data utilization, and a clearing house for continuous inspection of contract adherence. These are topics that have not yet been taken into discussion and investigation, due to the current progress of the PoC work. These are also topics that reveal their value only when the federated data spaces being piloted are large enough, and thus wait for large-scale piloting. It will also be interesting to see in future research how cross-sectoral use cases will change the value perception of federated data spaces. In the case of manufacturing, this can, for example, mean data-level collaboration and interoperability with energy, mobility (logistics), and finance sectors. It will also be crucial in the future to shift the product-level thinking from a linear build−use−discard model towards different circular economy cycles, and federated data spaces can provide a crucial basis for building these cycles.

## 5. Conclusions

Our early-phase exploration in federated data spaces within sustainable manufacturing revealed that the companies in question were actively seeking solutions for data sharing from several business objectives. Key drivers for this development according to the interviews were technological development at large, global megatrends such as sustainability, and their associated current/upcoming regulation, as well as transformations in leadership and working culture. This creates a sweet spot, where nascent business interest of companies can be addressed with research-based offering in a shared-risk setting. We can thus state that the answer to RQ1 is that the strategic relevance is already high and perceived as growing further in the future. In the case of RQ2, there are differing perceptions towards federated data spaces. The principal company seems to be less interested in a solution based on open data, while technology providers and suppliers see more benefits in it. We view this through the overall organization of an innovation ecosystem, where a principal stands to gain more business benefits from a “walled garden” setting, while technology providers and suppliers see benefits through interoperability towards multiple clients. In the case of RQ3, this early exploration revealed that multiple factors appear challenging for federated data space adoption. One is the perceived business value, and the different viewpoints based on roles within an individual ecosystem. Another is the maturity level of IDS software components and the lack of large-scale industry adoption at this point in time. Third is the capability of companies of different sizes within the ecosystem to carry out a necessary IT integration project that IDS adoption requires. This has also been clearly visible in the first empirical IDS PoC, from which the lessons are listed in Section 3.2.

This research should be considered as a status report of an ongoing industry−research collaboration project, with more results to be published in later articles. The findings serve both industry and academia in terms of an increased understanding and roadmapping towards the future. In the categorization of Jacobides et al. [2], our focus ecosystem represents both the business ecosystem with its focal company, as well as a platform ecosystem with a federated data space acting as a data sharing platform. These theoretical connections, coupled together with the necessitation of ecosystemic collaboration and continuous innovation in the face of sustainability challenges, as indicated by Nuutinen et al. [5], give further validation to our focus ecosystem as being representative of the future where digital and sustainable manufacturing is headed.

Building a larger-scale awareness and interest towards IDS architecture and data sharing in general requires inspiring use cases and company testimonials, especially in terms of sustainability and competitiveness. There is also a clear need for as-a-service solutions so that companies of different sizes and specializations can easily come onboard the federated data spaces [11]. In general, additional research is needed, from actors in different nations as well as international players, and across different levels of awareness and readiness, to gain an understanding of where we are in the state-of-practice and how data sharing and re-use are shaping up in Europe.

## Figures and Tables

**Figure 1 sensors-23-04315-f001:**
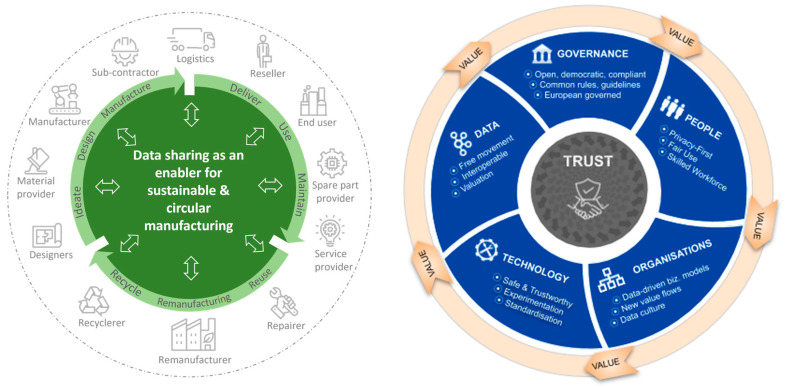
(**Left**) B2B data sharing acts as a key enabler in sustainable and circular manufacturing visions [7]. (**Right**) Value from data results from trusted collaboration [8].

**Figure 2 sensors-23-04315-f002:**
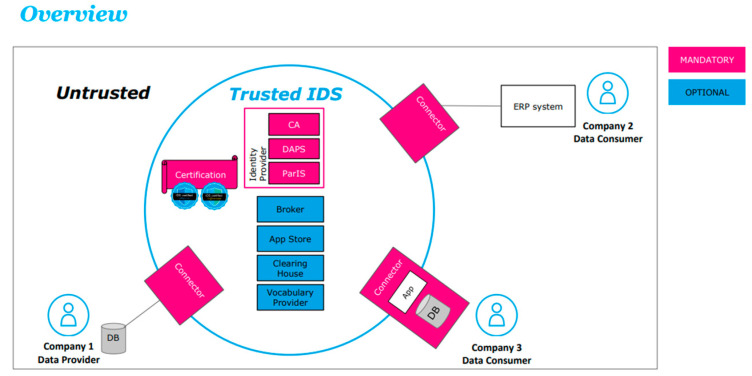
IDS high-level architecture [23].

**Figure 3 sensors-23-04315-f003:**
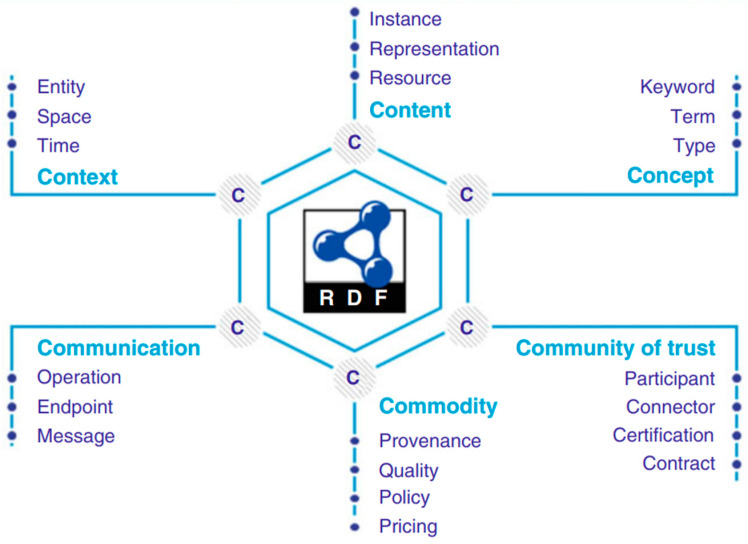
Concern hexagon illustrating the different dimensions that make up the domain-agnostic base-level data model of IDS [23].

**Figure 4 sensors-23-04315-f004:**
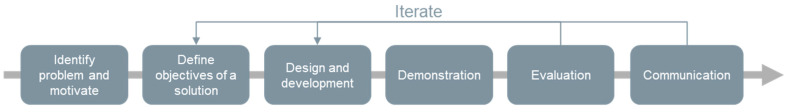
Design science research process [24].

**Figure 5 sensors-23-04315-f005:**
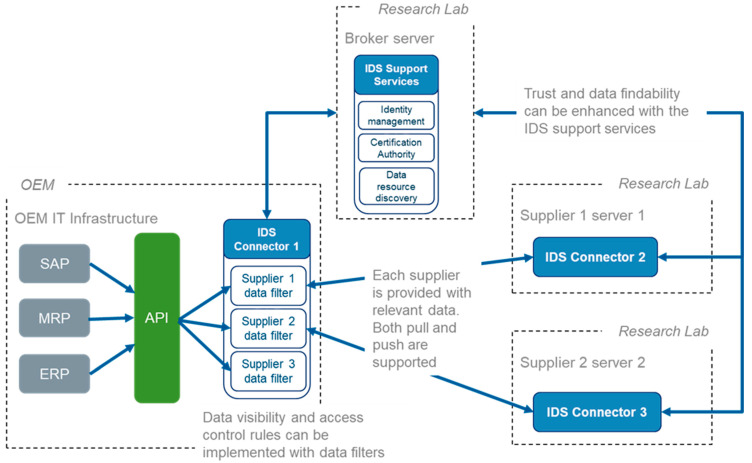
Architecture of the first data sharing PoC based on IDS components.

**Figure 6 sensors-23-04315-f006:**
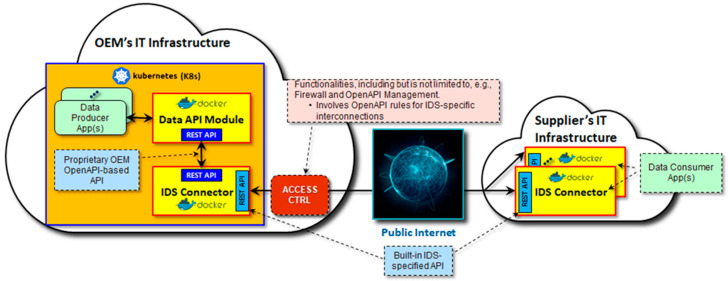
IDS connector adaptation in API management in the OEM IT infrastructure. The supplier-side IT infrastructure was emulated in the research partner lab at this stage.

**Table 1 sensors-23-04315-t001:** Interviewees from our target ecosystem.

Interviewee ID	Company Role in the Ecosystem	Interviewee Role in the Company
[I.1]	Principal	Team leader, analytics-centric team, and operational excellence group
[I.2]	Technology provider	Vice president and digital marketing
[I.3]	Technology provider	Technical director
[I.4]	Principal	Chief enterprise architect
[I.5]	Supplier	Sales development director

## Data Availability

The anonymized interview data are available from the authors upon request. The research diary is not anonymized and thus cannot be considered open data.

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
