# Peer review of "Exploring the Role of Federated Data Spaces in Implementing Twin Transition within Manufacturing Ecosystems"

_sensors, 2023, doi:10.3390/s23094315_

Round 1

Reviewer 1 Report

It would be better if the necessity related to Data Sharing Platforms were explained from the Life Cycle Management point of view.

Furthermore, it will be useful for the readers to have a summarized table or diagram which emphasizes the need to go for shared data platforms of manufacturing ecosystems.

Could have been used more relevant literature to reinforce the literature review

The interview guide used during the research is not presented, it would be better to present it for the benefit of the readers.

Furthermore, adequate emphasis has not been given to the interoperability and ethical concerns of data sharing.

Author Response

Please see the attached PDF for our reviewer response.
